# Attitudes and Perceptions of School Principals about the Contribution of Evaluation to the Efficient Operation of Schools Both at the Administrative and Educational Levels

Vasiliki Brinia [1] , Christos Katsionis [2,*] , Andriani Gkouma [2] and Ioannis Vrekousis [1]

[1] School of Humanities, Hellenic Open University, 263 35 Patras, Greece
[2] Teacher Education Program, Department of Informatics, Athens University of Economics and Business, 104 34 Athens, Greece
* Correspondence: chr.katsionis@gmail.com

**Abstract:** Evaluating teachers and educational work is now mandatory in the Greek educational system. The current study aims at investigating the attitudes and perceptions of principals of Piraeus Secondary Education units regarding the contributions of this evaluation law to the orderly functioning of schools in terms of administrative and educational aspects. For this qualitative research, a semi-structured interview was chosen as the results collection method. The interview questions included respondents' demographic information and interview questions corresponding to three thematic axes according to the research questions. The convenience sample consisted of twelve (12) principals of secondary school units from the greater Piraeus area. All interviewed principals recognized the need for the evaluation process in education and agreed on the importance of their roles and responsibilities for the effectiveness of the proper functioning of their units. Nine respondents underlined the need for training teachers on new technologies, six considered internal evaluation reports as a means of weaknesses identification and improvement and five believed the evaluation imposed by the Ministry of Education creates negative feelings. As concluded, evaluation is of great educational importance, as it contributes to teachers' professional development and schools' educational quality improvement. However, when implemented suddenly or imposed by the state, it causes pressure, stress, fear, nervousness, and insecurity.

**Keywords:** evaluation in education; internal evaluation; self-evaluation; educational work; administrative work; educational quality

## 1. Introduction

### 1.1. A Successful Evaluation System Leads to Quality Educational Work

Education constitutes a very important and essential social function that contributes to the progress of each individual and society. Educational systems nowadays are called to respond to a high level of complex changes to keep pace with current socio-economic developments [1]. At the same time, globalization, combined with rising competitiveness and the growth of knowledge societies, is shaping a new landscape in education, which needs to adapt its role in the current era [2]. The current trend argues that the effort to improve the quality of educational work must go through the implementation of a successful evaluation system [3]. The integration of assessment in education should not only focus on the student, but the educational system and the educational project should itself be evaluated [4]. In other words, "*evaluation in education is closely related to the whole of all functions of the educational system and includes curricula, textbooks, educational equipment, training of educational workers and of course the teacher and the school unit itself*" [5]. The teachers' and educational work's evaluation is the most fundamental issue for both the whole teachers' community and those who design educational policies. Questions such as "*when and how should teachers be evaluated?*", "*by whom should teachers be evaluated?*", "*what*

*kind of strategies should be implemented in order for teachers to improve their teaching?"*, are important topics of the debate on evaluation in Greece [6,7]. Many researchers express the opinion that evaluation can become a feedback mechanism in education and contribute to the reform of educational and administrative work [8–11].

*1.2. School Evaluation Policies in the International Context*

Teacher evaluation and educational policy implementation have been heavily researched worldwide and have evolved over the years with many countries witnessing regulation changes regarding internal or self-evaluation. An interesting case of this is the English system which has been focused on the recognition of the necessity of internal or self-evaluation instead of using the external model for a quality indicator [12]. Important research published in 2016 [13] between twenty-eight countries in the "OECD Review on Evaluation and Assessment Frameworks for Improving School Outcomes (2013)", pointed out that twenty-two countries had adopted either national or state-level policy frameworks regarding teacher evaluation and six countries were in the process of locally designing and implementing practices of feedback provision on educational work. We should always consider that each school unit has different needs, and the criteria of evaluation may differ based on them. It is also crucial for evaluation policies to be differentiated according to teachers' experience [14] and to deliver a culture of professional development. Principals play a major role in teacher evaluation and development, especially for early-career teachers [15], as they provide concrete feedback and are of additional value.

However, principal reports are not sufficient to assess teacher quality [16]. The educational standards and goals of each school unit should be clearly defined so that both teachers and principals are well-informed and identify areas for growth and development [17]. School autonomy in teacher evaluation in countries such as the Netherlands [18] contributes in many cases to students' autonomy and better performance. Autonomy and the selection of evaluation criteria are thoroughly examined. Since 2009, the U.S. public education system has implemented "high-stakes teacher evaluation systems" that had unexpectedly null effects on student achievement and attainment [19]. A healthy teacher evaluation that improves practice is due to be examined [20] with many questions raised on policy design and implementation.

Teaching is constantly reshaping, and teacher evaluation has been integrated into new educational policies and programs [21]. Many different examples of teacher evaluation practices have been applied across the globe, with the state playing a crucial role not only at a federal level, in countries such as England and Scotland, but also at a local level, in countries such as Australia, Canada and the United States.

*1.3. The Introduction and Implementation of Internal/Self-Evaluation Law in Greek Schools*

Expanding the discussion on the role of the state in the production, implementation and monitoring of teacher evaluation policies, recently a new legislative bill entitled "Collective planning, internal and external evaluation of school units in terms of their educational work" has been submitted in Greece, through the Ministry Decision 108906/DG4 of 10.09.2021, Government Gazette 4189/B/10-9-2021. According to the Ministry of Education [22], the legislative bill will be the means through which the teaching methods used by teachers, as well as the competence and effectiveness of the work carried out by educational staff, will be subject to assessment. The proponents of the new evaluation law aim at creating a better school for students. They believe that introducing assessments in schools will ensure students are provided with all the essential skills needed for their present and future. The new assessment is based on four pillars: (a) autonomy of school units, (b) introduction of teacher evaluation as a technique to strengthen and empower teachers, (c) improvement of all levels of education resulting in the achievement of an autonomous school, and (d) establishment of a system of church education with the main purpose of upgrading the clergy educational system under the supervision of the state.

The introduction of internal/self-evaluation in Greek school units, to transform schools into learning organizations, aims at contributing to improving quality in the areas of three functions: (a) pedagogical-learning function, (b) administrative function, and (c) teacher professional development function. It is the school itself that will design the actions and implement them as part of the internal evaluation process.

The Three Stages of the Process

According to the law, the process of internal/self-evaluation should be applied by all school units of Primary and Secondary Education every year. The process is governed by three stages:

First stage: Annual planning of the work of the teachers at the school unit. At the beginning of each school year, the principal of each school unit calls a meeting of the Teachers' Association to plan and schedule the collective actions to be carried out during the current school year. Teachers are divided into action groups and each group chooses a thematic axis from the nine defined by the Ministry of Education and selects actions that are considered most appropriate to improve respective indicators of that axis. The meeting may be attended, upon invitation, by other persons, such as the Education Advisor for Pedagogical Responsibility, staff members of Multidisciplinary Assessment and Counselling Centers and researchers from Research Centers, as well as scientists belonging to other institutions of the Ministry of Education.

Second stage: Implementation concerns the execution of the action plans created. When problems arise that make it difficult to carry out a planned action, discussion follows among the members of the action team, and reviews are made plus any changes if deemed necessary to achieve the action. The progress of the implementation of the initial planning of all actions is discussed at regular Teachers' Association meetings convened during the school year by the school principal; at least once every two months.

Third stage: Evaluation of the action plans implemented throughout the school year is based on the degree of achievement of the objectives set by the action groups at the beginning of the school year. At a special meeting of the Teachers' Association held at the end of the school year, the objectives fulfilled by the actions taken are discussed in relation to the improvement of the educational work and the functioning of the school unit. The school principal, in collaboration with the Teachers' Association, prepares the annual Internal Evaluation Report of the school unit. A summary of this report is then posted on the special digital application of the Institute of Educational Policy (I.E.P.).

The responsibility for the planning, implementation and evaluation of all collective actions that take place within the framework of internal/self-evaluation lie with each school's principal. During the evaluation of the actions, the work achieved in the school unit is assessed, per axis, and rated on a four-point scale. The points entered are from number one to number four and their significance are as follows [23]:

- 1 = Unsatisfactory operation. Many points need to be improved.
- 2 = Satisfactory operation. Some points need to be corrected.
- 3 = Good operation. Very few points need to be improved.
- 4 = Excellent operation.

While assessing the work carried out in the school unit to improve its operation at an administrative and educational level, the specific characteristics of the school unit, such as its infrastructure, the region in which it is located and its social characteristics, are also considered. The scoring is conducted by considering the three main functions of the school unit mentioned before. The pedagogical and learning function includes five sub-strands: (a) Teaching, learning, and assessment, (b) School dropout-attendance, (c) Relationships among students, (d) Relationships between students and teachers, and (e) School-family relations. The administrative function focuses on two sub-strands: (a) Leadership-organization and administration of the school unit, and (b) School and community. The professional development function focuses on two sub-axes: (a) Teachers' participation in training activities, and (b) Teachers' participation in national and European programs.

It should also be noted that each axis is described by appropriate indicators that characterize each of them. A summary of the Internal Evaluation Report is posted on the school's website. The documents related to the actions taken in the school within the framework of the evaluation process by the action teams are kept in a file kept at the school.

### 1.4. External Evaluation Process in Greek Schools

The external evaluation of school units is carried out by the Coordinators of Educational Work of Pedagogical Responsibility (CEWs), who prepare two reports.

The CEWs, after reading the internal evaluation reports of each school unit in their area of responsibility, prepare an External Evaluation Report for each school, which is posted on I.E.P. digital application each year. The report is accessible to the Teachers' Association of the school unit and the relevant Regional Center for Educational Planning and includes:

(a)   Observations and suggestions for the correction and improvement of the action plans implemented in the school unit within the framework of the collective planning evaluation.

(b)   The functions of the school unit, where they are evaluated with justification by thematic axis and rated on a ten-point scale.

(c)   Overall account of the strengths and weaknesses assessment of the school unit in relation to its three (3) core functions. The following are recommendations that will lead to the progress of the school unit, after considering other factors such as the specific operating conditions, the social characteristics of the pupils and the initiatives of the school teachers.

The Report on the External Evaluation of schools under the responsibility of CEWs is written by the CEWs for all the school units under their responsibility and uploaded on the I.E.P. digital application each year. In this report, after indicating the strengths and weaknesses of the school units and making an evaluation of all the actions in all the school units, opinions and suggestions are made on teachers' training to improve their educational work, with the ultimate purpose of improving the functioning of the school units. All the good practices are also mentioned and the issues that should be further explored, as well as the problems that have arisen and the ways they can be addressed.

Except for the above, we should also focus on the new Greek Law 4823 entitled *"Evaluation of the Work of Executives, Teachers and other Specialist Staff of Public Primary and Secondary Education"* and submitted through Government Gazette 136/A/03-08-2021. Principals' leadership under teacher evaluation reform has been studied across many countries [24,25]. According to that, evaluation should be carried out in accordance with the principles of impartiality, service, educational and support capacity and efficiency of staff, while involving teachers and members of specialized teaching and support staff, and while considering the specificities of the educational, pedagogical and support work provided [26]. Participation in any evaluation process is compulsory for all and is an official duty of the utmost importance, given the objectives it pursues. Failure to carry out the evaluation process by an individual or by certain groups of school units and other educational or support structures constitutes specific disciplinary misconduct. Especially for an education executive who refuses to take part in the evaluation process, either as evaluator or as the person being evaluated, then this official should not only be replaced and lose the position they have held until then, but also be excluded from the selection process for the next eight (8) years for filling any position of education executive [27].

### 1.5. Purpose and Research Questions of Present Study

Thus, the present study aims at making an original contribution towards recording the attitudes and perceptions of principals of secondary schools regarding the evaluation that will be implemented in schools during the school year 2021–2022. We aim at identifying and presenting the main points of the new legal framework for the implementation of the evaluation/self-evaluation of school units, as well as examining the role of the school unit management in the implementation of internal evaluations. Alongside those considerations,

the main objectives constitute the investigation of the factors that correlate evaluation with the improvement of the quality of educational work and the identification of the elements that link the quality of the school unit's operation with the pedagogical and administrative work. The contribution of the study, to the extent that it is relevant, is crucial to the improvement of the evaluation process since the law will be implemented from this school year. Finally, our goal is to highlight the criteria and the selection process of school principals in secondary education through the current study.

To achieve these objectives, we present the following research questions:

1.  What are the attitudes and perceptions of the principals of secondary schools about evaluation in education and how do they perceive their role as principals in this process according to the new law?
2.  What is the opinion of secondary school principals on the evaluation of the school unit and the imprint that the internal evaluation reports will leave on the schools?
3.  What is the opinion of the principals of secondary school units regarding the criteria and the selection process for their selection as principals, according to the new evaluation law?

## 2. Materials and Methods

### 2.1. Method

To conduct this study, qualitative research was chosen. Ensuring the validity and reliability of qualitative research is achieved by specific research consistency criteria and through the thorough description of the research process followed [28]. The choice of qualitative research aims at understanding and drawing conclusions through the opinions, sayings and perceptions of the participants in this research. According to Patton, with qualitative research we do not predict what may happen in the future. It is simply an attempt to understand the current situation experienced by the participants, thus aiming at understanding important information [29]. More specifically, this study was conducted through semi-structured interviews. With respect to other research methods, the main advantage of the semi-structured interview is that it gives the possibility of an in-depth study on the issue under consideration, because researchers are given the opportunity to modify, add or analyze the questions, depending on the development of the interview, in order to obtain all the information they really need [30].

Regarding the research consistency criteria mentioned above, we ensured that the validity and reliability criteria proposed by Guba and Lincoln [31] were met, as follows:

#### 2.1.1. Reliability of the Research

The decision to investigate the specific research questions in the school context of secondary education was not a random decision, but was taken to satisfy the reliability criterion of prolonged engagement with the research field. The researchers have had a considerable educational experience in both pedagogy and teaching.

#### 2.1.2. Validity of the Research

The validity of the research was achieved through disciplined self-checking and self-awareness of the researchers during the collection, analysis and presentation of the research data. The researchers tried to present the procedures followed in an accurate way.

#### 2.1.3. Trustworthiness of the Research

Apart from reliability and validity, according to Shenton (2004), to ensure trustworthiness in qualitative research there are four specific aspects that need to be taken into consideration: (a) credibility, (b) transferability, (c) dependability, (d) confirmability [32]. Those have been applied to our study as follows:

Credibility of the Research

The researchers were "external observers", with no involvement in affecting the opinions of the sample, remaining neutral and maintaining, at the same time, their internal observers' role due to their own teacher status. The random sampling of participants also contributed to this factor.

Transferability of the Research

Transferability of the research is related to the extent to which our findings can be applied to other situations. Even if the results extracted by qualitative research can be quite specific, each unique case can be an example in a wider universe of similar situations. In our study, we have examined a small group of people, whose characteristics can be applied to a wider population of principals in Greece and abroad — if a similar survey could be situated in an international context — as well as the questions asked, and the interview flow followed, were not specifically formed for those people and could be used under other circumstances.

Dependability of the Research

Dependability involves utilizing techniques that demonstrate that if the study were to be replicated under the same conditions, using the same methods and participants, similar outcomes would be achieved. That is why we have tried to provide as comprehensive and detailed as possible a description of the research processes and its action items, so that our approach can be considered as a "prototype" by a future researcher who should also evaluate the degree to which proper research procedures were followed.

Confirmability of the Research

In qualitative research, the concept of confirmability is analogous to objectivity. Even if it is challenging to ensure real objectivity, the researchers have minimized the influence of their personal characteristics and preferences on the findings, by letting the interviewees share their unfiltered thoughts. All the interviews have been recorded, so that participants' arguments can be efficiently documented and through coding matched to the relevant thematic questions. That way, any additional human interpretation has been minimized on our side.

2.1.4. Ethics

Last but not least, regarding the ethics aspect of our research, participants have been fully informed about the purpose of the study, have agreed to the recording of the interviews and to the way their content will be used. As well, they have given their consent for the publication of this research by a specific form that we have shared with them.

*2.2. Sample*

Before proceeding to the presentation and analysis of the findings, it is worth highlighting some qualitative characteristics of the population that took part in the research. The research sample consisted of twelve (12) principals serving in secondary education schools in the greater Piraeus area who had been chosen through the random convenience sampling method. As Shenton (2004) states by summarizing relevant studies, random sampling of individuals to serve as informants limits the researchers' bias in selecting participants and ensures that this group of people will be representative of a larger population [32]. Some of the interviews were conducted in person, following an expression of willingness by the principals, while others were conducted via the internet, after sending the survey questionnaire, to give principals the necessary time to prepare their answers.

Noteworthy is that seven out of the twelve principals were male while the majority of them (N = 8) were aged between 50 and 59 years old. Regarding their teaching experience, most of the participants (N = 7) were serving over 25 years in education, while the sample members' experience as directors ranged from 1 to 15 years. The education level of the

principals has also been examined in terms of their potential postgraduate studies, and finally it is worth highlighting that the expertise of the principals interviewed ranged among many different subjects such as literature, mathematics, physics, etc.

The following Table (Table 1) shows all the demographic data of the sample.

**Table 1.** Sample Demographics.

| Variable | Options | f | % |
|---|---|---|---|
| Gender | Male | 7 | 58.3 |
| | Female | 5 | 41.7 |
| Teaching Experience | Less than 15 years | 0 | 0 |
| | 15–25 years | 5 | 41.7 |
| | Over 25 years | 7 | 58.3 |
| Principal Experience | 1–4 years | 5 | 41.7 |
| | 5–8 years | 3 | 25.0 |
| | 9–12 years | 3 | 25.0 |
| | Over 12 | 1 | 8.3 |
| Education Level | Bachelor's degree | 4 | 33.3 |
| | Postgraduate degree | 8 | 66.7 |
| | Theology | 1 | 8.3 |
| | Greek Literature | 1 | 8.3 |
| | Mathematics | 1 | 8.3 |
| | Physics | 1 | 8.3 |
| Field of expertise | English Literature | 1 | 8.3 |
| | French literature | 1 | 8.3 |
| | Physical Education | 2 | 16.7 |
| | IT | 3 | 25.0 |
| | Geography | 1 | 8.3 |

*2.3. Data Analysis*

To create the interview questions, we first made a list of specific items that we wanted to include in the survey results. The interview questions included in Part A are the demographic data of the respondents and in Part B are the interview questions regarding the three thematic axes of the research questions. The questions were selected by the researchers based on the thematic content of the research and their personal judgment.

Firstly, a consent form for participation in the research was sent to the principals who wished to be interviewed, informing them that only their demographic data would be used for statistical data processing. The questionnaire was then sent out so that the respondents would have time to prepare themselves for the answers they would give during the interview, as well as to take some notes to discuss with the researcher. Then, the interview appointment would follow.

The interviews were conducted between 27 March and 19 April 2022, and their duration ranged from 12.5 min at the shortest to 24 min at the longest. Despite the disadvantage that several interviews were not conducted face-to-face but rather via the internet, due to the ease of the survey and the COVID-19 pandemic, they were nevertheless conducted in a pleasant and warm atmosphere without any problems.

All principals who were interviewed were made aware of the importance and purposes of the research and were assured of their anonymity. They were also asked for permission to record the interview so that it could be transcribed into text to clarify points that were not well understood on first listening to an interview [33]. There was a climate of mutual trust, and assurance was given that they would have access to the results if they requested it.

The thematic analysis method was used to analyze the responses from the interviews. The following steps were followed to apply this method:

(a) First, the recorded interviews were transcribed into text format and a simple notation system was used.

(b)    Then, after several readings of the recorded interviews, all the views expressed in the answers given by the principals to each question individually were recorded.

(c)    Commonalities in the responses to each interview question given individually were then identified.

(d)    Finally, for the common responses given to the interview questions, their frequency was measured, and the interview extracts were separated according to the research questions.

## 3. Results

### 3.1. "What Are the Attitudes and Perceptions of the Principals of Secondary Schools about Evaluation in Education and How Do They Perceive Their Role as Principals in This Process According to the New Law?"

Regarding the First Research Question, "*What are the attitudes and perceptions of the principals of secondary schools about evaluation in education and how do they perceive their role as principals in this process according to the new law?*", principals have been asked four different questions through the questionnaire.

First Question: "*Do you believe that Evaluation in Education is a necessary process? Give the main reasons supporting your opinion.*"

In the first part of the question, ten out of the twelve principals (83.3%) answered that evaluation is useful and constructive so it should clearly be a necessary process for education. One Principal (8.33%) believed that evaluation is a tool that only if used properly by the state will bring improvement in schools, while one principal (8.33%) was simply in favor of evaluation because as he said, "*evaluation can offer several things to both the school and the teacher*".

On the second part of the question, there were many different answers, and some of them should be highlighted. Ten principals (83.3%) agreed that it would help teachers to become better, and therefore help them in their professional development. It will motivate those teachers who have been inactive until now, it will make many others more active, more proactive, and more efficient, so that they will take more initiative, improve their teaching style and the way they carry out their work and their general duties. Nine out of twelve principals (75%) argued that evaluation will contribute to improving the learning and pedagogical functioning of schools, while four of those nine principals especially believed that evaluation will document the weaknesses and weak points that exist in schools, so that we can look at them and try to improve them, so that teaching can become more efficient and, in general, progress can be made in all areas in the school. They also had the belief that with evaluation, cooperation and co-responsibility between teachers in school will become more developed.

On the other hand, five principals out of twelve (41.7%) believed that the specific evaluation that the Ministry of Education wants to implement from the 2021–2022 school year creates fear, anxiety and insecurity among teachers, especially about what will follow the evaluation. The same number of principals (N = 5) were also of the opinion that there are teachers who believe that after the evaluation there will be punitive procedures with penalties, salary reduction and even dismissals. For this reason, they stressed that an evaluation should not be punitive in nature, because then it is sure to fail. Finally, one of those five principals expressed the view that precisely because evaluation creates many problems in its implementation, the way in which it is carried out should be changed if it is to contribute to the improvement of schools.

Second Question: "*What is your opinion on the evaluation of the educational work in your school unit? What factors do you think influence your work as an evaluator?*"

Regarding the next question, most of the principals (N = 7, 58.3%) replied that this should be performed by considering only the educational work that each teacher performs in the school, i.e., it should be on their pedagogical status. Half of the principals argued that the relationships that each teacher has developed with other colleagues and with the pupils in the school will also play a decisive role, while five principals reported that for the evaluation they would consider the teacher's overall behavior, i.e., the willingness they show in the school, the consistency of their school timetable, whether they

are consistent in all their obligations in the school, etc. Finally, one principal said that they would also emphasize the relationships that each teacher in the school has with the parents of the students.

More specifically, about the factors that will influence their work as evaluators, four out of twelve principals expressed the opinion that to have good judgment towards their colleagues, the evaluation they will be given should not be punitive. The same number of participants argued that to some extent the friendly relations that have been developed between some teachers and the principal of the school unit will affect the evaluation of their school unit and create problems. They even stressed that problems would also be created by the bad relations that a principal is likely to have with some of their colleagues, who in this case could use the evaluation as a means of blackmail. Two principals out of twelve (16.7%) stressed that they would also be affected by health problems or other serious problems that each teacher, either themselves or their family, might have, since, as they said, "*each principal and each director is first and foremost a human being and sympathizes with them*".

Third Question: "*Do you believe that your responsibilities and your role as principal are decisive factors for the effectiveness and quality of the educational work in the school unit under your responsibility? Please give the reasons that support your opinion.*"

In the first part of this question, half of the participants believed that a principal's responsibilities and role are decisive factors for the effectiveness and quality of the educational work in the school unit. Five principals (41.67%) claimed that their responsibilities and role are very important factors in the effectiveness and quality of educational work in the school unit under their responsibility, but not the determining factor, since there are other factors that play a role, so obviously effectiveness and quality in the school unit comes as a combination of many factors. Finally, one principal (8.33%) argued that, with the responsibilities that they have, they cannot do the things they want in the school of their responsibility, and therefore the effectiveness and quality of the educational work in the school unit depends much more on the intentions of the teachers' association in the school.

Regarding some specific reasons, eleven out of the twelve principals (91.7%) were of the opinion that the school principal is the guide and encourager for their colleagues, the one who advises them and makes them feel confident, the one who communicates and cooperates with everyone and the one who will solve any problems that arise in the school without tensions and conflicts, in a climate of mutual respect and dialogue. Seven principals especially believed that the principal plays the most decisive role in creating a positive and pedagogical climate in the school among teachers, between teachers and students and between teachers and parents. For this reason, "*principals must be fair with everyone, be flexible, be adaptable and treat everyone equally well and kindly.*" Finally, one principal expressed a quite interesting opinion when they argued that actually, a principal does not have the powers that they should have, and who pointed out that "*The principal is the last among equals. He is the one who bears all the responsibilities in everything and as a result a scapegoat. Their actions depend on the composition of the faculty association, the spirit of cohesion it has, and the mood it has. Because if the vision that the principal wants to fulfill for their school is not shared by the rest of the teachers in the school or they don't want to share it, then nothing will happen.*"

Fourth Question: "*What incentives could be given to the educational community to improve their professional skills?*"

Regarding the fourth question, all principals believed that a very important motivation for teachers to improve their professional skills is the training and seminars that should be given to teachers. The principals' feedback on the issues that training should cover, and the ways and the times they should be implemented, was quite interesting.

First, about those issues, nine principals (75%) claimed that teachers need training on new technologies. There are several teachers in schools who are aged over fifty years old and most of them are not well digitally educated. The same was the case with younger teachers, whose kind of expertise is not directly related to IT, such as philologists, theologians, language teachers, etc. As an example, some principals mentioned that "*there are*

*teachers who do not know what a 'mouse' is*". Two principals argued that teachers also need training on the cognitive part of the subject they teach, which should be substantial and more practical, i.e., not on a theoretical level but actually helping the teacher to make the lesson more meaningful and better in their classroom. The same number of participants argued that training should be given to teachers in the pedagogical part as well, as they said that "*sterile knowledge alone is not enough for a teacher to do their teaching properly. There are many teachers who 'suffer' in the pedagogical part of their work at school in which they need help*".

Second, with regards to the ways and the times in which the training of teachers should be implemented, four out of twelve principals (33.3%) believed that training should be compulsory for all teachers, especially trainings on new technologies and the innovative ways of teaching shaped by new technologies. More specifically, one principal said that: "*It is wrong for the state to leave each teacher with a choice in training because there will be some teachers who will not be trained at all while others, who will be the same all the time every time, will attend them. This will create a gap between those who attend training and those who do not. And as time goes on, this gap will continually grow.*" Three principals also felt that training can be held outside of the teacher's school hours, in the evenings or on weekends, especially when it is possible to do them online. The same number of principals, however, are of the opinion that training should take place within the teacher's school hours, that they should take place at regular intervals, and that the teachers attending them should be released from their teaching duties or have reduced hours to be actively involved in some way. Since teachers also have their personal lives and their daily schedules and their families, they cannot be forced to attend training seminars in the evenings or on weekends.

*3.2. " What Is the Opinion of Secondary School Principals on the Evaluation of the School Unit and the Imprint That the Internal Evaluation Reports Will Leave on the Schools?"*

Regarding the Second Research Question, "*What is the opinion of secondary school principals on the evaluation of the school unit and the imprint that the internal evaluation reports will leave on the schools?*" principals had been asked three different questions through the questionnaire.

First Question: "*What in your opinion should be the criteria that would be most appropriate for the evaluation of the school unit?*"

According to the answers given to this question, ten principals (83.3%) believed that a very important criterion, if not the most important, is the climate in the school, as well as the proper cooperation of all the actors that make up the school unit, i.e., the principal, the teachers and the students at the school. Five principals were of the opinion that the dropout of pupils during the school year is also a criterion suitable for evaluating the school unit. They stressed, however, that student drop-out rates should not be compared between different schools and especially between schools located in different areas. Two of the principals were of the opinion that criteria for the evaluation of school units should also be: (a) The particularities of each school unit, and (b) the support available for children with learning needs and the school's participation in training programs (environmental, health education, European, etc.).

Second Question: "*Do you believe that the effectiveness of educational work is a function of the quality of education of the school unit?*"

About this question, all twelve principals agreed that the effectiveness of the educational work is indeed a function of the quality of education of the school unit, since obviously the better the quality of the education provided to the teachers in the schools, the more effective the teachers are in their work. They also claimed that poorly provided education cannot be effective. On the other hand, however, they believed that the effectiveness of educational work does not only depend on the quality of education provided, but also on other factors such as: (a) The physical infrastructure of each school, (b) the gaps in the number of teachers in schools and the time needed to fill these gaps, (c) the relationships between teachers and pupils in the school, and (d) tutorials and private lessons that students may take outside the school.

Third Question: "*What in your opinion is the imprint that the Internal Evaluation Reports will leave and how could they be evidence of quality for the school unit?*"

In this question, eight out of twelve principals (66.7%) believed that internal evaluation reports will provide a basis for contributing to school improvement and upgrading. Half of the principals were of the opinion that the internal evaluation reports will help find out where the school units are lacking, to know their weaknesses and to try to improve them.

Four principals claimed that they are in favor of evaluation because they believe that internal evaluation reports will bring about some changes for the benefit of the school. One of them stated that "*Even a poor implementation of evaluation in schools is better than no evaluation at all.*" To show the great importance of evaluation in education, another principal pointed out that: "*The internal evaluation report should be like the compass on a ship. Without a compass, the ship does not know where it is going. The same is the case in education where without the internal evaluation report we are not able to know what is produced and how it is produced*".

*3.3. "What Is the Opinion of the Principals of Secondary School Units Regarding the Criteria and the Selection Process for Their Selection as Principals, According to the New Evaluation Law?"*

Regarding the Third Research Question, "*What is the opinion of the principals of secondary school units regarding the criteria and the selection process for their selection as principals, according to the new evaluation law?*", principals had been asked two different questions through the questionnaire.

First Question: "*Who in your opinion would be the appropriate bodies-groups for the evaluation of the school principal?*"

According to the answers given to this question, nine out of twelve principals (75%) were of the opinion that a suitable person for the evaluation of the school principal is the school's pedagogical responsibility advisor. They claimed that the pedagogical responsibility consultant visits the school on a regular basis, is in contact with the school principal, and works with the principal to solve the serious problems that plague the school, so they know what is happening in each school under their responsibility and they know how the principal of each school works. Alongside those observations, eight out of twelve principals believed that an appropriate group to evaluate the school principal is the school's faculty association. In fact, five of them believed that the teachers' association is the one that should have the first and last say and that its opinion should count more than the other evaluators. They claimed that the teachers in the school are the ones who know better than anyone else what is happening in the school they work in, since they are in the school daily. Three out of twelve principals argued that parents of the pupils should also have an opinion on their evaluation, especially regarding the school's cooperation with them. Finally, one principal, expressed the view that the local community should be involved in the evaluation of the school principal too.

Second Question: "*What do you think should be the main criteria in the selection process of principals in education?*"

In response to this question, all twelve principals were of the opinion that a key criterion should be the teacher's years of service, i.e., teaching experience in the school. Nine participants stated that a teacher's master's or doctoral degree should also play a role, while eight out of twelve principals argued that a very important criterion should also be the teacher's knowledge of new technologies. About the personality of the principal, seven out of twelve (58.3%) argued that one criterion should be the character of the teacher, since as they pointed out: "*They should be fair, treat all the teachers in the school equally and have harmonious relations with them, be cooperative, willing to contribute, work hard, be able to handle difficult situations and resolve any problems, tensions or crises that arise in the school.*" Three of the principals argued that one extra criterion should be the interview, which would reveal more elements of the candidate's character and personality and will show whether the candidate has the appropriate leadership qualities needed. Finally, seven of the principals also believed that it would be helpful and appropriate for a teacher before

becoming a principal to have served as an assistant principal so that they will have gained administrative experience that would help them in their future responsibilities as principals.

## 4. Discussion

Moving to the discussion section, our study aims to answer the three research questions already stated before and to examine their impact in the context of the existing literature.

Regarding the first part of the First Research Question, all twelve interviewed principals responded that evaluation is a process that should exist in education, either because they believe that it is a necessary process, or because they think that it is useful for teachers and for the school, or because they are of the opinion that it will eventually bring about an improvement in the quality of school functioning in its educational and administrative work. It will also develop collaboration among teachers and generally help them in their professional development, which is in full agreement with Sowell's research [25]. It will contribute to a better pedagogical and administrative functioning of schools, since it will enable us to identify the weak points that exist in schools so as to make efforts to improve them, by giving teachers the chance of further personal development [15]. These responses are in line with the research by Koutouzis, who argues that through evaluation the school unit becomes a "living learning organisation" where its members are constantly cultivating new, innovative, and expanded forms of thinking and action, with the aim of how to learn together [34].

However, they expressed some objections to the specific law that the Ministry of Education wants to implement, regarding its contribution to the improvement of the quality of the operation of school units. This perception exists because they see how most teachers have accepted this evaluation as an additional forced work and bureaucratic process, which will take valuable time away from teaching, which is their main role. In addition, principals are of the opinion that this evaluation creates a feeling of fear, anxiety and insecurity among teachers because they do not know what will follow after their own evaluation as well as after the evaluation of their school unit. This view is in line with research by Athanasiou and Georgousi who argue that evaluation causes pressure, anxiety and nervousness in teachers resulting in a decrease in their performance [9]. Nevertheless, this should not be perceived as a threat by school personnel, but rather as constructive feedback they will need for self-improvement and eventually for higher educational services.

The second part of the First Research Question examined how the school principals of secondary education perceive their role as evaluators in the schools under their responsibility. Regarding the evaluation of their fellow teachers in the school unit of their responsibility, they believe that the main criteria are their teaching skills and scientific qualifications, their initiatives in terms of improving the school unit, their consistency in the school timetable and the degree of achievement of the additional tasks assigned to them since the beginning of the school year. These criteria are fully in line with those highlighted in Lazaridou's study [35]. Additionally, the respondents argued that another important criterion is the good relations that teachers have with each other, as well as with the students of the school. Furthermore, they emphasized that their assessment will be influenced by the level of interpersonal relationships they have developed with their fellow teachers. It is apparent that the human element plays an important role in each teacher's evaluation by the relevant director. This is a risk that inevitably has to be taken in the process of evaluation, as the director is the most qualified person and has an overall view of the work conducted throughout the academic year. At the same time, all the respondents agreed that their responsibilities and role as principals are very important to decisive for the effectiveness and quality of the educational work in their unit of responsibility, since the principal is the one who has a vision for their school and tries to achieve it by creating a positive pedagogical climate. In other words, they believe that the role of the principal in the school is catalytic, in line with the findings of many studies [10,18,34,36]

Regarding the incentives that could be offered to the teaching community to improve their professional skills, all the participants of the survey believe that teacher training

is definitely a very basic incentive for teachers to improve the quality of the education they provide. This view is in line with the results of several studies already mentioned before [11,37,38]. Most principals focused on teacher training in new technologies, since they stressed that many teachers, especially those over fifty years of age, are quite behind in this area. Technology is gaining ground in every aspect of our daily lives and education cannot be excluded. The recent health crisis due to COVID-19 demonstrated that the use of new technologies is deemed necessary for a modern educational system. Another point of argument appeared to be the timeframe where this training will take place, with the participants being split among those who suggest that training should take place within their working hours or outside of those. Finally, they expressed the view that financial incentives should be provided on a permanent basis. These are in line with the results of research by Lochmiller and Mancinelli who state that it is necessary to support teachers through appropriate training programs, as well as to provide incentives and proper and timely information on evaluation issues [24].

Moving to the Second Research Question, a large percentage of principals believe that the criteria for the evaluation of the school unit should be the climate that prevails in the school and the proper cooperation of all the school unit's stakeholders, namely the principal, teachers, students, parents and the municipality. This is in line with the findings of the study by Katsarou and Dedouli and the study by Lazaridou [35,39]. To a lesser extent, they consider that the following should also be taken into account in these criteria: the drop-out rate of pupils during the school year, the support provided by the school to pupils with learning needs, the integration or reception sections in the school, the school's participation in all kinds of programs, the particularities of the school (such as the area where it is located and the social background of the area), and the image that the school presents to the local community. In essence, a school should be considered as a vital part of society where the citizens of the future are nourished.

However, they believe that the effectiveness of the educational work is related to factors that are not directly linked with the quality of the education offered by teachers and which can instead be attributed to the school infrastructure and the manning of the school. Lastly, another important factor has to do with the mental connection between students and teachers and the way they trigger their interest and inspire them to be more active in the learning process.

Regarding the second part of the Second Research Question, most of the principals are of the opinion that internal evaluation reports will contribute to the improvement and upgrading of the quality of the operation of the school units in the educational and administrative work, because with these reports the objectives set at the beginning of the school year will be reviewed and could be further modified or enriched if the need arises. Such reports could serve us as "checkpoints" that show the way to better quality in education. These findings are in full agreement with those of the study by Iordanidis and Tsagalidou and the study by Georgoudakis [40,41]. However, there were also some principals who were skeptical and expressed the view that because evaluation in education started to be implemented from this school year, they were not sure about the effect that the internal evaluation reports would bring about, and whether they would contribute to the improvement of the educational and administrative work provided in schools. There was also the view that these reports may create problems between schools or between teachers in schools due to the inequalities they may bring about and, more generally, may have a negative impact on relations between teachers and on the school climate. This is in line with the study by Ahanasiou and Georgousi and the study by Harisis [9,42]. Finally, there was also the view that evaluation reports will go unnoticed and will not be dealt with.

Concerning the Third Research Question, the participants in the survey believe that appropriate bodies or groups for the evaluation of school principals should be: the school's pedagogical responsibility advisor; the school's teachers' association, as well as, to a lesser extent, the association of parents and guardians of the school's pupils; the municipal education department with the school committee; the school's pupils; and the social

stakeholders. However, opposing opinions were also expressed by a few principals who believe that the evaluation of school principals should be carried out by external evaluators to ensure the objectivity of the judgement.

Finally, regarding the second part of the Third Research Question, concerning the criteria that a teacher should have in the selection process for a school principal, all the principals interviewed are of the opinion that one of the main criteria should be the years of service of the teacher in secondary schools, because they consider that a school principal must have a great deal of teaching experience as well as experience from the extra-curricular activities that are carried out in schools. They also believe that a master's or doctoral degree should be a selection criterion, but they do not specify whether this degree should be relevant to educational leadership and administration. In contrast, in his study, Mavrogiorgos advocates the distinction of postgraduate degrees and argues that those with relevance to educational leadership and management should take precedence [43]. In addition, the respondents believe that it is good to have served in the position of assistant principal in a school unit so as to have gained administrative experience as well. They further stressed that, apart from the above "formal qualifications", a very important criterion in the selection process of a teacher for head of a school unit should be their character as well as the leadership qualities they possess, because the unit head should have a well-rounded personality as the one who will inspire and guide all the other teachers in the school and the one who will create the right climate for the safe and proper functioning of the school unit. This is in line with the research by Kelesidis who states that the selection of education managers should move away from a consolidated and formal process with questionable criteria [44].

## 5. Conclusions

In conclusion, evaluation and self-evaluation are crucial factors for schools to properly work and provide all their members (i.e., students, teachers, principals) with great educational and learning opportunities as well as a positive environment, which our literature review on an international level has also pointed out. In our research, we aimed at investigating principals' perceptions about the new law for evaluation in Greek secondary schools in terms of these factors: (a) their role in the evaluation process, (b) the imprint that the internal evaluation reports could have on schools, and (c) the criteria and the selection process for their selection as principals, according to the new evaluation law. According to the results of the present study, we are given the opportunity to suggest that a meaningful dialogue between teachers and the state should take place regarding the main axes of development of the institution of evaluation to achieve consensus on the part of teachers for the implementation of the law of evaluation in schools, so as not to end up with evaluation as a formal and meaningless bureaucratic process.

Additionally, any evaluation law should create a climate of security, and not uncertainty, among teachers so as to be adopted by them and successfully implemented. The state should institutionalize seminars per subject area that will enable continuous training for teachers, with emphasis on new technologies and modern teaching methods, since according to this study and the educational policies in the international context, teachers need training so that they can develop themselves. The existence of a national strategic plan for the evaluation of teachers and educational units that has the broad consensus of the country's political forces will ensure the continued implementation of evaluation in schools regardless of any changes in political power in the country.

Finally, it should be highlighted that this research was conducted in the hope that it will be the impetus for further study in the field of teacher evaluation and self-evaluation of school units, which are a prerequisite for the modernization and improvement of education in Greece.

## 6. Limitations of Present Study and Suggestions for Further Research

Beyond the useful implications that our research points out, there were a few limitations which should be highlighted as well. The main limitations of this research are related to the period when it was conducted, since several interviews were not conducted face-to-face but rather via the internet, due to the ease of the survey and the pandemic of COVID-19. One more limitation is the small sample size of only twelve (12) principals as well as the local status of the survey, since all the principals interviewed are serving in the Piraeus area of Attica-Greece. Situating the research in such a context in combination with the convenient sample selection methodology, limits at some point the generalizability of the findings for a larger number of principals/schools.

Regarding any suggestions for future research, there is a need for further investigation of the same study with a larger sample population and on a larger, even Panhellenic scale, because we believe that more reliable results will be obtained this way. Moreover, the same research should be conducted, not only at the level of principals, but also at the level of secondary school teachers. Furthermore, it is important to repeat a similar study after a reasonable period of time since the implementation of the evaluation law in the schools studied, so that a comparison can be made between the conclusions before and after its implementation in schools, in order to determine to what extent the implementation of the evaluation law has brought about the expected results in education and to note the weaknesses and shortcomings of the bill in order to bring about appropriate improvements.

**Author Contributions:** Conceptualization, V.B. and I.V.; methodology, V.B., C.K. and I.V.; software, I.V.; validation, V.B. and I.V.; formal Analysis, C.K. and I.V.; investigation, A.G. and I.V.; resources, C.K. and A.G.; data curation, C.K. and I.V.; writing—original draft preparation, C.K., A.G. and I.V.; writing—review and editing, V.B., C.K., A.G. and I.V.; visualization, C.K.; supervision, V.B.; project administration, V.B.; All authors have read and agreed to the published version of the manuscript.

**Funding:** This research received no external funding.

**Institutional Review Board Statement:** Ethical review and approval were waived for this study because it wasn't relevant to the type of research conducted.

**Informed Consent Statement:** Written informed consent has been obtained from the survey participants to publish this paper.

**Data Availability Statement:** The data presented in this study are available on request from the corresponding author. The data are not publicly available due to confidentiality and anonymity of research participants.

**Conflicts of Interest:** The authors declare no conflict of interest.

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
