# Peer review of "Attitudes and Perceptions of School Principals about the Contribution of Evaluation to the Efficient Operation of Schools Both at the Administrative and Educational Levels"

_education, doi:10.3390/educsci13040366_

Round 1
Reviewer 1 Report (Previous Reviewer 3)
The article presents significant improvements, namely regarding the literature review, with the incorporation of state of art regarding evaluation policies and global trends, and the improvement of the methodological part, through clarification.
Nevertheless, it could have a more critical reading that strengthens the arguments and academic soundness of final considerations of the article.
Author Response
Dear Reviewer 1,
Thank you for taking the time to review our manuscript and for your valuable feedback! Here is a point-by-point response to your suggestions.
1) The article presents significant improvements, namely regarding the literature review, with the incorporation of state of art regarding evaluation policies and global trends, and the improvement of the methodological part, through clarification.
Answer: Thank you for your kind words
2) Nevertheless, it could have a more critical reading that strengthens the arguments and academic soundness of final considerations of the article.
Answer: Thank you for your feedback. We have extended our argumentation in the "Discussion" section, so that the implications of our study can be further highlighted.
Reviewer 2 Report (New Reviewer)
1. I think authors have done a very good job in writing their work. Introduction has been established to captures the reader's attention and interest, provide readers with an understanding of the subject matter, and summarize the central argument of the work. However, if it possible, please add headings or sub-heading to better the introduction. So that, comfort to read the paper will ease readers. You can do like what you did for the method section.
2. It is not wrong to say “validity and reliability” in qualitative research term. However, the term “trustworthiness” can be best used. I can not ask you to change, just a suggestion to improve the paper.
3. “…hey have been chosen through the 250 random convenience sampling method”.. I think authors should further detail the statement, perhaps could insert citations on the sampling approach.
4. Findings and discussion have been well presented. Findings have provided a synopsis of the results followed by an explanation of key findings and presented results with overall highlights on primary purpose of the research questions.
Discussion is also recommended describing, analyzing, and interpreting the qualitative findings, informing the significance of the results and relating to the prior studies .
Author Response
Dear Reviewer 2,
Thank you for taking the time to review our manuscript and for your valuable feedback! Here is a point-by-point response to your suggestions.
1) I think authors have done a very good job in writing their work. Introduction has been established to captures the reader's attention and interest, provide readers with an understanding of the subject matter, and summarize the central argument of the work. However, if it possible, please add headings or sub-heading to better the introduction. So that, comfort to read the paper will ease readers. You can do like what you did for the method section.
Answer: Thank you for your kind words. We have added suitable sub-headings to the “Introduction” section, so that it is easier for any reader to follow the flow of the section and the argumentation stated.
2) It is not wrong to say “validity and reliability” in qualitative research term. However, the term “trustworthiness” can be best used. I can not ask you to change, just a suggestion to improve the paper.
Answer: Thank you for your feedback. It’s an accurate note on the “Methodology” section structure. According to your comment and based on Academic Editor’s notes too, we have maintained the terms "reliability" and "validity", while at the same time, we have added the trustworthiness aspect of our research and elaborated on its 4 factors.
3.“…hey have been chosen through the 250 random convenience sampling method”.. I think authors should further detail the statement, perhaps could insert citations on the sampling approach.
Answer: Thank you for your feedback. We have added a new reference regarding the random sampling method, which is also related to the credibility aspect of our research's trustworthiness as described in the "Method" subsection in "Methodology.
4. Findings and discussion have been well presented. Findings have provided a synopsis of the results followed by an explanation of key findings and presented results with overall highlights on primary purpose of the research questions.
Discussion is also recommended describing, analyzing, and interpreting the qualitative findings, informing the significance of the results and relating to the prior studies.
Answer: Thank you for your feedback. We have analyzed in more detail the "Discussion" section, by enhancing the argumentation on the results so that we further highlight their importance and possible implications.
This manuscript is a resubmission of an earlier submission. The following is a list of the peer review reports and author responses from that submission.
Round 1
Reviewer 1 Report
Dear authors,
The paper deals with an interesting topic, however, I have some suggestions related to the content and the consistency of the paper that I find necessary to address:
- Reading the article, I feel like a Literature Review chapter is missing. In the Introduction chapter, you made some references to the literature, but I would have loved to see more research on the subject instead of reading about the way the process of evaluation is being structured in Greece, more so as the topic does not reference this later on.
- Pay more attention to how you present the information. In Chapter 2 Materials and Methods, e.g., in line 206 you say that the teaching experience for 7 out of the 12 interviewed principles is over 26 years, yet the classification in the table states over 25 years. The way you have presented it now leaves year 26 away.
- Subchapters from Chapter 3 Results are not appropriate “1st Research Question”. It would have been better to put the actual research question there, especially since you have sub-questions for every research question. One other thing that I do not find necessary is classifying the answers, e.g., in lines 255-271 you state that "8 out of the 12 interviewed principles consider evaluation clearly a necessary process for education" and then you continue saying "two principals (16.67%) believed that it is useful and constructive, so it should be present in education", which basically means the same thing so this classification is not really useful.
- Chapter 4 Discussions is repeating the information from Chapter 3 Results, but it is better structured as it connects the results with the findings from the present literature. My suggestion is to leave just one chapter called Results and Discussions.
- The order of chapter Conclusions and Limitations should be switched. First, you conclude then you present the limitations and future research perspectives.
Best regards,
Author Response
Dear Reviewer 1,
Thank you for taking the time to review our manuscript and for your valuable feedback! Here is a point-by-point response to your suggestions.
1) Reading the article, I feel like a Literature Review chapter is missing. In the Introduction chapter, you made some references to the literature, but I would have loved to see more research on the subject instead of reading about the way the process of evaluation is being structured in Greece, more so as the topic does not reference this later on.
- Thank you for your feedback. By following the Journal’s guidelines for the structure of the article we have added the context and the aim of our study as well as the literature review around the subject in the “Introduction” chapter. Based on your suggestions, we have added some more research in the “Introduction” about the international context concerning evaluation policies in education, by citing relevant references.
2) Pay more attention to how you present the information. In Chapter 2 Materials and Methods, e.g., in line 206 you say that the teaching experience for 7 out of the 12 interviewed principles is over 26 years, yet the classification in the table states over 25 years. The way you have presented it now leaves year 26 away.
- Thank you for your feedback. We have revised the way we present the qualitative characteristics of the population that took part in our research and especially the age classifications.
3) Subchapters from Chapter 3 Results are not appropriate “1st Research Question”. It would have been better to put the actual research question there, especially since you have sub-questions for every research question. One other thing that I do not find necessary is classifying the answers, e.g., in lines 255-271 you state that "8 out of the 12 interviewed principles consider evaluation clearly a necessary process for education" and then you continue saying "two principals (16.67%) believed that it is useful and constructive, so it should be present in education", which basically means the same thing so this classification is not really useful.
Thank you for your feedback. First of all, we have revised the subchapters' titles (e.g. “1st Research Question”) and have replaced them with the actual research question respectively. Secondly, we have revised our Results presentation and have merged any classifications that seemed unnecessary because of expressing almost the same thing.
4) Chapter 4 Discussions is repeating the information from Chapter 3 Results, but it is better structured as it connects the results with the findings from the present literature. My suggestion is to leave just one chapter called Results and Discussions.
- Thank you for your feedback. We have revised both Chapters so that we present both clarified and detailed findings in the “Results” Chapter as well as minimize the repeated information in the “Discussion”. That’s why, by following the Journal’s guidelines for the structure of the article, we kept both Chapters distinguishable, so as we present the findings in the "Results" one, by adding participants’ testimonials as well, and on the other hand, include the key points of the results with their connection with the present literature in the “Discussion” Chapter.
5) The order of chapter Conclusions and Limitations should be switched. First, you conclude then you present the limitations and future research perspectives.
- Thank you for your feedback. We have switched the order of the two chapters.
Best regards,
The Authors
Reviewer 2 Report
The article deals with a very specific issue regarding evaluation in Greek schools. It is systematic in nature, with a clear goal and aplicable research methods. It can be improved, if necessary, by adding an international perspective and comparing different approaches for evaluation.
Author Response
Dear Reviewer 2,
Thank you for taking the time to review our manuscript and for your valuable feedback! Here is a point-by-point response to your comments.
1) The article deals with a very specific issue regarding evaluation in Greek schools. It is systematic in nature, with a clear goal and applicable research methods.
- Thank you for your kind words.
2) It can be improved, if necessary, by adding an international perspective and comparing different approaches for evaluation.
- Thank you for your feedback. Based on your suggestions, we have added some more research in the “Introduction” section about the international context concerning evaluation policies in education, by citing relevant references.
Kind regards,
The Authors
Reviewer 3 Report
The paper presented, starting from an important theme, namely in the current context of the Greek educational system, deserved some improvements regarding the theoretical and empirical framework in the study area. It was essential to strengthen it by presenting studies on teacher evaluation and evaluation policies in an international context. In particular, considering that this study was conducted in the wake of a recently approved legal framework in Greece concerning teacher evaluation, it would be essential to refer to the international context concerning evaluation policies in education.
Author Response
Dear Reviewer 3,
Thank you for taking the time to review our manuscript and for your valuable feedback! Here is a point-by-point response to your comments.
1) The paper presented, starting from an important theme, namely in the current context of the Greek educational system, deserved some improvements regarding the theoretical and empirical framework in the study area. It was essential to strengthen it by presenting studies on teacher evaluation and evaluation policies in an international context. In particular, considering that this study was conducted in the wake of a recently approved legal framework in Greece concerning teacher evaluation, it would be essential to refer to the international context concerning evaluation policies in education.
- Thank you for your feedback. The paper has been focused on the current context of the Greek educational system, since the legal framework for evaluation is quite recent, however, following your suggestions we’ve expanded the theoretical framework of our study and connected that with the situation described in the greek context. More specifically, we have added some more research concerning evaluation policies at an international level, especially in education systems that are quite different from the greek one, by citing relevant references both in the "Introduction" section as well as in the "Discussion" one.
Kind regards,
The Authors